# Electroencephalography as a diagnostic tool for late-onset efavirenz neurotoxicity syndrome

Sam Nightingale[1,2]*, Salvatore Ssemmanda[1], Lawrence M. Tucker[1,2], Roland W. Eastman[1,2], Eddy B. Lee Pan[1,2]

**1** Neurology Department, Groote Schuur Hospital, Cape Town, South Africa, **2** Neuroscience Institute, University of Cape Town, Cape Town, South Africa

* sam.nightingale@uct.ac.za

## Abstract

### Introduction

To examine electroencephalogram (EEG) as a diagnostic tool for late-onset efavirenz (EFV) neurotoxicity syndrome (LENS), an uncommon but severe and potentially fatal complication of EFV therapy.

### Methods

We conducted a Retrospective case-control study. EEGs from confirmed cases of LENS (clinical syndrome and plasma EFV >4ug/mL) recorded from June 2016 to May 2021 were compared with control EEGs from the same time-period. Controls were adults (18–70 years) with a similar indication for EEG (eg. encephalopathy or confusion), dysrhythmia generalised grade II, and LENS excluded. EEGs were reviewed by two blinded interpreters given a description of the characteristic EEG changes, ie. persistent, diffuse, high voltage, bisynchronous, monomorphic 4–7 Hz theta frequency waveforms with transient attenuation on eye opening. Interpreters were asked to determine whether EEGs showed definite, probable or no changes.

### Results

Thirteen LENS cases were compared with 50 control EEGs. Interpreter 1 labelled 11/13 LENS cases as having define or probable changes, and interpreter 2 labelled 10/13. Interpreter 1 labelled probable changes in 1/50 controls and interpreter 2 in 3/50. Neither interpreter labelled any controls as having definite changes. Interrater reliability was good with 95% agreement and a Cohen's kappa of 0.83. Sensitivity of EEG under these conditions for the diagnosis of LENS was 85% and 77% for interpreters 1 and 2 respectively, and specificity was 98% and 94%.

**Data Availability Statement:** All relevant data are within the paper and its Supporting Information files.

**Funding:** The authors received no specific funding for this work.

**Competing interests:** The authors have declared that no competing interests exist.

## Conclusions

EEG is a useful tool in the diagnosis of LENS which can be used to aid clinical decisions while awaiting EFV levels, or in low-resource settings where EFV levels are not available.

## Introduction

Efavirenz (EFV) is a non-nucleoside reverse transcriptase inhibitor, which until recently was recommended in the first line antiretroviral treatment regimen for HIV by the World Health Organisation [1]. EFV remains widely prescribed globally due to its effectiveness and high barrier to resistance.

EFV has been associated with neurotoxic side effects. This typically manifests as nightmares and anxiety occurring in the weeks following commencement of the drug. These are usually self-limiting and do not require discontinuation of EFV unless severe. In the longer term, EFV has been associated with mild cognitive impairment, depression and suicidality [2, 3].

The late-onset efavirenz neurotoxicity syndrome (LENS) is a severe, uncommon side effect of efavirenz (EFV). Two case series from South Africa describe a syndrome of ataxia and progressive encephalopathy associated with supratherapeutic EFV levels [4, 5]. To date LENS has not been described outside the African continent. Affected patients are typically females with low body weight, often in association with concomitant isoniazid use for tuberculosis prophylaxis. Without discontinuation of EFV, encephalopathy can progress to coma and the condition can be fatal. With prompt discontinuation of EFV, prognosis is generally good [4–6].

More recently the pharmacogenomics of LENS has been described [6]. CYP2B6 is the main route of EFV metabolism, with CYP2A6 a lesser pathway [7]. those with genotypically slow CYP2B6 metaboliser status, the CYP2A6 becomes the main route. Isoniazid is an inhibitor of CYP2A6, providing a double hit in those with genotypically slow CYP2B6 receiving this drug [4, 8]. This is a common scenario in southern Africa where TB is endemic. The South African HIV Clinicians Society guidelines recommend that isoniazid preventive therapy should be started at antiretroviral initiation or added to the treatment regimen of patients already on antiretrovirals who have not yet received isoniazid preventive therapy, once active TB has been excluded [9]. In a recent study, all patients with LENS had slow CYP2B6 metaboliser status and 13/15 (87%) had concomitant isoniazid administration [6].

It is crucial that clinicians are able to make a prompt diagnosis of LENS to prevent progression to poor outcomes and death. Currently diagnosis is made by supratherapeutic EFV levels in the context of an appropriate clinical syndrome [4–6]. EFV concentrations are measured by mass spectrometry, and as such are not available in many of the settings in which LENS occurs. Where EFV levels are available, results can take several weeks to return. The clinical syndrome alone can be suggestive of LENS, but there is a broad differential for encephalopathy in people living with HIV. Misdiagnosing LENS—for example in someone with a central nervous system opportunistic infection–could lead to inappropriate and unnecessary changes to antiretroviral therapy. As such, adjunctive tools are required to aid the diagnosis of LENS, which are widely available and provide a rapid result.

At our tertiary level neurology unit at Groote Schuur Hospital in Cape Town, South Africa, we have received multiple cases of LENS. Most patients received electroencephalogram (EEG) as part of diagnostic workup for encephalopathy. We noticed a distinct EEG pattern which appeared to be characteristic of LENS (Fig 1) [4]. Here we describe these changes in detail and report a case-control study to examine the effectiveness of EEG in the diagnosis of LENS.

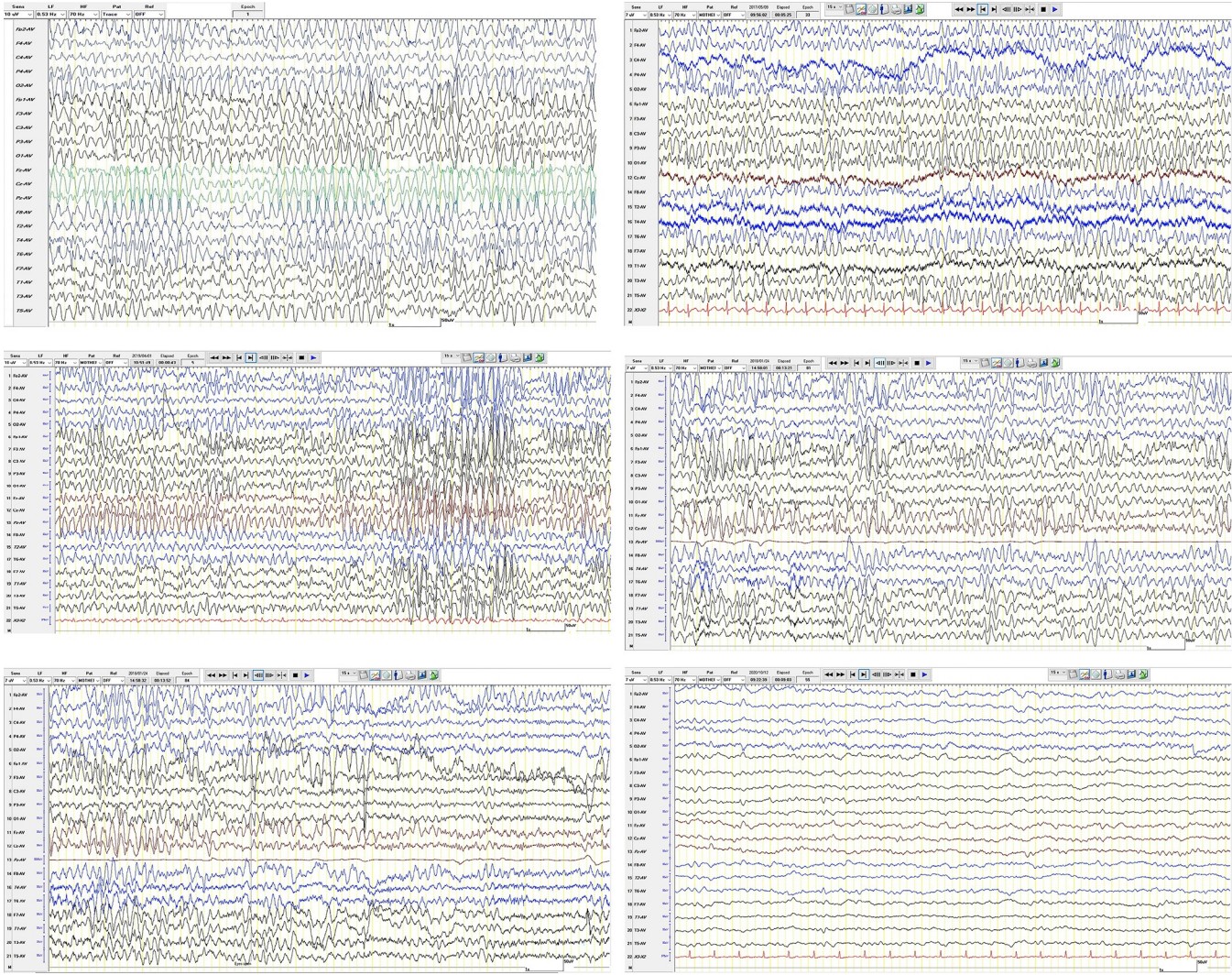

**Fig 1. Characteristic EEG changes of LENS.** a-e are 5 EEG epochs from 4 LENS cases showing persistent, diffuse, high voltage, bisynchronous, monomorphic 4–7 Hz theta frequency waveforms. Transient attenuation on eye opening is shown in e (same case as d). f is a non-LENS control for comparison—a 44 year-old male living with HIV with disseminated TB and acute hydrocephalus. EEG shows some low amplitude semi-rhythmic theta but in contrast to LENS this is intermittent and varying, mixed with low amplitude fast frequencies and some low amplitude delta transients. 3 further examples from non-LENS controls are given in supplementary material.

## Methods

### Population

LENS cases were identified from laboratory records of supratherapeutic EFV levels from June 2016 to May 2021. Case notes were examined to confirm a clinical diagnosis of LENS. Case definition for LENS was an appropriate clinical syndrome with supratherapeutic EFV levels (>4ug/mL) and available EEG recording.

Controls were identified from adults (18–70 years) with EEG electronic records over the same 5-year period. Control definition was: (i) indication for EEG any one of: encephalopathy, encephalitis, delirium, confusion, neurocognitive decline, or suspected EFV toxicity (chosen to be similar to the clinical presentation of LENS), (ii) EEG showing dysrhythmia generalised, grade II (see below), (iii) not receiving EFV, or receiving EFV but toxicity excluded with non-

toxic plasma EFV levels (<4ug/mL) and/or an alternative diagnosis. Those with unknown HIV status were excluded due to uncertainty over EFV exposure.

The study was approved by the Human Research Ethics committee (HREC 258/2022) of the University of Cape Town (UCT) and the Institutional review board of Groote Schuur Hospital. Approval was granted to record anonymised data without specific consent given the retrospective nature of the study.

## EEG

Scalp EEGs are performed by qualified neuro-clinical technologists using a Nihon Kohden EEG machine and software version II, alongside a modified international 10–20 electrode placement system. EEGs were recorded for a minimum of 20 minutes during which activation procedures including hyperventilation, intermittent photic stimulation, and arousal procedures for mental alerting are performed. Hyperventilation was omitted during the COVID-19 pandemic.

EEGs were classified according to a modified Mayo Clinic system [10]. We restricted EEGs to those with dysrhythmia generalised, grade II (moderate to severe) as this corresponded to the changes observed in LENS. Those with no changes (normal) or mild abnormality (grade I) were excluded, as were those with more specific abnormalities and seizure activity (grade III).

## Blinded EEG interpretation

EEGs from cases and controls were assessed by two blinded interpreters (clinical neurologists experienced in the interpretation of EEG). EEGs were presented in a random order with all clinical details removed. Both interpreters were provided with a single EEG recording for reference representing the typical changes of LENS (Fig 1A). They were informed that these changes were persistent, diffuse, high voltage, bisynchronous, monomorphic 4–7 Hz theta frequency waveforms with transient attenuation on eye opening. They were asked to review all EEGs and record whether they felt the recording showed characteristic changes. If they felt the characteristic features were unequivocally present, they were asked to record as 'definite'. If some changes were present, but they were equivocal, they were asked to label the recording 'probable'. If no changes were present, they labelled the recording 'no changes'.

## Statistical analysis

Statistical analyses were performed using GraphPad Prism version 9. Descriptive variables between cases and controls were compared with Student's t-test and Fisher's exact. The sensitivity, specificity, positive and negative predicative value of EEG to diagnose LENS under these conditions were calculated for each interpreter, with combined definite and probable changes representing a positive result. Inter-rater agreement was calculated by Cohen's Kappa.

## Results

13 cases of LENS were identified with supratherapeutic plasma EFV levels (>4 ug/mL). EFV levels were exceeded 20 ug/mL (the upper limit of quantification in our lab) in 12/13 cases, with one case having levels of 10.4 ug/mL.

Controls were identified as follows. 6254 EEGs were recorded over the 5-year period, 5306 of which were graded as dysrhythmia generalised, grade II. 3581 were excluded as no indication for EEG was recorded. 1447 were excluded as the recorded indication did not match our control definition. Further records were excluded due to age (n = 27), duplicate records (n = 65), and unknown HIV status (n = 134). Two were receiving EFV but did not have EFV

**Table 1. Characteristics of cases versus controls.** EFV = Efavirenz, N/A = not applicable, EEG = encephalogram.

| | LENS cases (n = 13) | Controls (n = 50) | p value |
|---|---|---|---|
| Age, mean (SD) | 33.2 (7.12) | 45.5 (12.0) | 0.0008 |
| Female, n (%) | 12 (92%) | 28, (56%) | 0.022 |
| HIV Positive, n (%) | 13 (100%) | 14 (28%) | N/A |
| Efavirenz exposed, n (%) | 13 (100%) | 8 (16%) | N/A |
| **Self-identified ethnicity, n (%)** | | | |
| Black | 12, (92.3%) | 21 (42%) | 0.002 |
| Coloured | 0 (0%) | 24 (48%) | |
| White | 0 (0%) | 3 (6%) | |
| Unknown | 1/13 (7.7%) | 2 (4%) | |
| **Stated indication for EEG, n (%)** | | | |
| EFV neurotoxicity | 6, (46%) | 1 (2%) | <0.001 |
| Encephalopathy | 4, (31%) | 37 (74%) | |
| Encephalitis | 0, (0%) | 5 (10%) | |
| Confusion | 0, (0%) | 3 (6%) | |
| Delirium | 0, (0%) | 3 (6%) | |
| Neurocognitive decline | 0, (0%) | 1 (2%) | |
| Subclinical seizures | 3, (23%) | 0 (0%) | |
| **EEG interpretation, n (%)** | | | |
| Interpreter 1—definite | 9 (69%) | 0 (0%) | N/A |
| Interpreter 1—probable | 2 (15%) | 1 (2%) | |
| Interpreter 1 –no changes | 2 (15%) | 49 (98%) | |
| Interpreter 2—definite | 6 (46%) | 0 (0%) | |
| Interpreter 2—probable | 4 (31%) | 3 (6%) | |
| Interpreter 2 –no changes | 3 (23%) | 47 (94%) | |

levels measured and did not have an alternative diagnosis, hence did not meet criteria for case or control. The remaining 50 EEG recordings formed our control group.

The characteristics of our 13 cases and 50 controls are given in the table. LENS cases were younger, more likely to be female, and more likely to be of black ethnicity. The majority of controls (74%) underwent EEG to investigate encephalopathy. In LENS cases EEG was performed for suspected EFV toxicity in under half (46%); the remainder underwent EEG for encephalopathy or the suspicion of underlying subclinical seizures.

Interpreter 1 labelled 11/13 LENS cases as having define or probable characteristic EEG changes, and interpreter 2 labelled 10/13. Interpreter 1 labelled probable changes in 1/50 controls and interpreter 2 in 3/50. Neither interpreter labelled any controls as having definite changes (Table 1). Interrater reliability was good with 95% agreement and a Cohen's kappa of 0.83 (95% CI 0.66–1.0).

The sensitivity of EEG for the diagnosis of LENS under these conditions was 85% and 77% for interpreters 1 and 2 respectively. Specificity was 98% and 94%. The positive predictive value was 92% and 77% for interpreters 1 and 2 respectively. The negative predictive values were 96% and 94%.

## Discussion

This study demonstrates that a characteristic EEG appearance reliably corresponds to a laboratory confirmed diagnosis of LENS, an uncommon but potentially fatal complication of EFV treatment. Accurate and prompt diagnosis of LENS is important so that EFV can be

substituted for an alternative antiretroviral drug. EEG is a low-cost tool which provides a rapid result allowing clinicians to make decisions in real time. This contrasts with the measurement of plasma EFV levels, which can take considerably longer (typically days to weeks depending on the lab) and are not available in many low-resource settings.

The changes described are different from other causes of drug-induced toxic encephalopathy, for example benzodiazepines or barbiturates (beta-rhythms), clozapine (generalised slow and spike wave discharge), metronidazole (frontal, monomorphic, sharply contoured theta activity), lithium (mixed theta and delta frequency with high voltage sharp delta wave bursts), and valproate induced hyperammonaemia (generalised slowing) [11–13]. In our sample the changes were not caused by HIV or by non-toxic EFV exposure, as the control group included 14 people living with HIV without LENS, the majority (8/14) of whom were receiving EFV treatment. The cause of the observed EEG appearance in LENS is not clear but may represent an effect of EFV on the oscillatory and spike rhythms of the reticular thalamic nuclei [14, 15].

Our centre has developed an online e-learning course to upskill clinicians and technicians in EEG interpretation (https://studyeegonline.com/), appropriate for low-resource settings. The course has been widely used globally, including in many southern African regions with high HIV prevalence. As such we believe that EEG, although not universally available, is an appropriate investigation for many diverse African settings.

It should be noted that the EEG changes were not universal in LENS and this test should not be used to rule out the condition. No EEG changes were identified by either interpreter for two LENS cases, despite EFV levels >20 ug/mL. The cause of this is unclear. We did not have sufficiently detailed clinical information to determine whether these participants had less severe disease. Our lab does not quantify levels above 20 ug/mL, hence it is possible the plasma EFV levels were lower in these cases compared to LENS cases with more pronounced EEG changes. Against this however, the one LENS case which had EFV levels in the quantifiable range (10.4 ug/mL) did have EEG changes (these were labelled 'probable' by both interpreters).

## Conclusions

We have demonstrated that EEG is a useful diagnostic tool with good sensitivity and specificity for the investigation of LENS. We suggest that, where available, EEG should form part of the diagnostic workup for this condition.

## Supporting information

**S1 Fig. Examples of EEGs from non-LENS control participants.** a. 58 year-old woman living with HIV. Diagnosis: Wernicke's encephalopathy. EEG shows a mixed background with large theta and delta frequencies, with superimposed semi-rhythmic theta and faster frequencies. In contract to LENS cases, the semi-rhythmic activity is unevenly distributed in time and space and generally low amplitude. b. 32 year-old HIV-negative female. Diagnosis: acute disseminated encephalomyelitis (ADEM). EEG was marred by excessive EMG artefacts. Epoch demonstrated has filtered the high frequencies to show the underlying brain activity, although some residual EMG attracts are still evident. There is intermittent theta but this contrasts to LENS cases due to the presence of a mixture of other frequencies with delta transients, fragments of alpha and also some beta activity. c. 47 year-old HIV-negative female. Diagnosis: hepatic encephalopathy. EEG shows quite rhythmic fast theta sinusoidal activity which fluctuates between low and very low amplitudes. Some low amplitude faster frequencies also present and occasional slower theta transients (the latter not shown in this epoch). Of the 4 control EEGs shown here, this would be the most similar to LENS cases, but lacks the very high

amplitudes associated with this condition.
(ZIP)

**S1 Data.**
(XLSX)

## Author Contributions

**Conceptualization:** Sam Nightingale, Eddy B. Lee Pan.

**Formal analysis:** Sam Nightingale, Salvatore Ssemmanda, Eddy B. Lee Pan.

**Investigation:** Lawrence M. Tucker, Roland W. Eastman.

**Methodology:** Sam Nightingale, Salvatore Ssemmanda, Lawrence M. Tucker, Roland W. Eastman, Eddy B. Lee Pan.

**Supervision:** Sam Nightingale, Eddy B. Lee Pan.

**Writing – original draft:** Sam Nightingale.

**Writing – review & editing:** Salvatore Ssemmanda, Lawrence M. Tucker, Roland W. Eastman, Eddy B. Lee Pan.

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
