## [Decision Letter · Decision Letter 0]

27 Mar 2023

PONE-D-23-04560Electroencephalography as a diagnostic tool for late-onset efavirenz neurotoxicity syndromePLOS ONE

Dear Dr. Nightingale,

Thank you for submitting your manuscript to PLOS ONE. After careful consideration, we feel that it has merit but does not fully meet PLOS ONE’s publication criteria as it currently stands. Therefore, we invite you to submit a revised version of the manuscript that addresses the points raised during the review process.

We look forward to receiving your revised manuscript.

Kind regards,

Tommaso Martino, M.D.

Academic Editor

PLOS ONE

Journal Requirements:

Additional Editor Comments:

Please provide more EEG traces. It would be useful for the reader if at least 3 EEG of patients affected by LENS are included in the paper, given the rarity of this disease. Inclusion of slight different EEG traces will improve the skills also of our readers.

As reference, provide at least 1 (and 3 more as supplementary material) EEG traces of control patients, describing their diagnosis and EEG findings.

Reviewers' comments:

Reviewer's Responses to Questions

**Comments to the Author**

1. Is the manuscript technically sound, and do the data support the conclusions?

Reviewer #1: Yes

Reviewer #2: Yes

2. Has the statistical analysis been performed appropriately and rigorously? 

Reviewer #1: Yes

Reviewer #2: Yes

3. Have the authors made all data underlying the findings in their manuscript fully available?

Reviewer #1: Yes

Reviewer #2: Yes

4. Is the manuscript presented in an intelligible fashion and written in standard English?

Reviewer #1: Yes

Reviewer #2: Yes

5. Review Comments to the Author

Reviewer #1: The authors examined EEG as a diagnostic tool for LENS. To do so, they conducted a retrospective case-control study including 13 LENS cases and 50 controls. Two blinded interpreters examined the EEGs. Interpreter 1 labeled LENS cases 11/13 as having definite or probable changes, while interpreter 2 labeled 10/13. Interpreter 1 labeled probable changes in controls 1/50 and interpreter 2 3/50. Interrater reliability was good and the sensitivity of the EEG for the diagnosis of LENS was around 80% and the specificity was over 90%. The authors conclude that EEG is a useful tool in the diagnosis of LENS.

I have no comments

Reviewer #2: This is a retrospective case-control study aimed to compare EEGs from 13 confirmed cases of late-onset

efavirenz (EFV) neurotoxicity syndrome (LENS) with 50 control EEGs recorded in adults with a similar indication for EEG, dysrhythmia generalised grade II, and without LENS. Interrater (1 and 2) reliability was adequate, sensitivity of EEG for the diagnosis of LENS was 85% and 77% for interpreters 1 and 2 respectively, and specificity was 98% and 94%. So the Authors demonstrated that EEG is a useful and cheap tool in the diagnosis of LENS, before to get the EFV level.

Data are clearly reported and discussed. Statistical analysis is adequate. The findings are relevant and helpful to address the daily clinical practice with patients presenting LENS.

6. PLOS authors have the option to publish the peer review history of their article (what does this mean?). If published, this will include your full peer review and any attached files.

Reviewer #1: **Yes: **Umberto Aguglia

Reviewer #2: No

---

## [Author Response · Author response to Decision Letter 0]

15 Jun 2023

We note the positive reviews and thank both the reviewers for their input. There were no specific issues to respond to. Editors suggestions addressed with EEGs added.

---

## [Editor Report · Decision Letter 1]

19 Jun 2023

Electroencephalography as a diagnostic tool for late-onset efavirenz neurotoxicity syndrome

PONE-D-23-04560R1

Dear Dr. Nightingale,

We’re pleased to inform you that your manuscript has been judged scientifically suitable for publication and will be formally accepted for publication once it meets all outstanding technical requirements.

Kind regards,

Tommaso Martino, M.D.

Academic Editor

PLOS ONE

---

## [Editor Report · Acceptance letter]

4 Jul 2023

PONE-D-23-04560R1 

Electroencephalography as a diagnostic tool for late-onset efavirenz neurotoxicity syndrome 

Dear Dr. Nightingale:

I'm pleased to inform you that your manuscript has been deemed suitable for publication in PLOS ONE. Congratulations! Your manuscript is now with our production department. 

Kind regards, 

on behalf of

Dr. Tommaso Martino 

Academic Editor

PLOS ONE